# Effects of Salinity and Rootstock on Nutrient Element Concentrations and Physiology in Own-Rooted or Grafted to 1103 P and 101-14 Mgt Rootstocks of Merlot and Cabernet Franc Grapevine Cultivars under Climate Change

Kleopatra-Eleni Nikolaou [1,*], Theocharis Chatzistathis [2], Serafeim Theocharis [1], Anagnostis Argiriou [3], Stefanos Koundouras [1] and Elefteria Zioziou [1]

1. School of Agriculture, Aristotle University of Thessaloniki, 54124 Thessaloniki, Greece; sertheo@agro.auth.gr (S.T.); skoundou@agro.auth.gr (S.K.); ezioziou@agro.auth.gr (E.Z.)
2. Hellenic Agricultural Organization (H.A.O.) 'Demeter', Institute of Soil and Water Resources, 57001 Thessaloniki, Greece; t.chatzistathis@swri.gr
3. Institute of Applied Bioscience 6th Km Charilaou–Thermi Road P.O. Box, 60361 Thessaloniki, Greece; argiriou@certh.gr
* Correspondence: nikolaouk@agro.auth.gr

**Abstract:** Under the current and future climate crisis, a significant rise in soil salinity will likely affect vine productivity in several Mediterranean regions. During the present research, the rootstock effects on salinity tolerance of Merlot and Cabernet Franc grapevine cultivars were studied. In a pot hydroponic culture, own-rooted Merlot and Cabernet Franc grapevine cultivars or grafted onto the rootstocks 1103 P and 101-14 Mgt were drip-irrigated with saline water. A completely randomized $3 \times 2 \times 2$ factorial experiment was designed with two vine rootstocks or own-rooted vines, two scion cultivars, and 100 mM NaCl salinity or control treatments, with six replications. A significant effect of scion cultivar, rootstock, and salinity was observed for most of the measured parameters. At the end of salinity stress period, leaf, shoot, root, and trunk nutrient concentrations were measured. Salinity stress increased Chloride (Cl) and Sodium (Na) concentrations in all parts of the vines and decreased leaf concentrations of Potassium (K), Calcium (Ca), Magnesium (Mg), Nitrogen (N), and Iron (Fe). In contrast, salinity stress increased leaf Boron (B) concentrations, whereas that of Manganese (Mn) remained unaffected. Leaf chlorophyll concentration decreased from 42% to 40% after thirty and sixty days of salt treatment, respectively. A similar trend was observed for the CCM-200 relative chlorophyll content. Salinity significantly decreased steam water potential (Ws), net $CO_2$ assimilation rate (A), and stomatal conductance ($g_s$) in all cases of grafted or own-rooted vines. Sixty days after the beginning of salt treatment, total Phenolics and PSII maximum quantum yield (Fv/Fm) decreased significantly. The rootstock 1103 P seems to be a better excluder for Na and Cl and more tolerant to salinity compared to 101-14 Mgt rootstock.

**Keywords:** grapevine; salinity; rootstocks; ion concentration; $CO_2$ assimilation; stomatal conductance; chlorophyll fluorescence

## 1. Introduction

It is well known that global climate models predict an increase in aridity in the next future. It has been reported that global aridity has increased substantially since the 1970s due to recent drying over Africa, southern Europe, East and South Asia, and eastern Australia [1]. Climate change, particularly temperature increase, is seen as a major challenge for crop production. The global mean temperature for 2020 (January to October) was $1.2 \pm 0.1$ °C above the 1850–1900 baseline, used as an approximation of pre-industrial levels [2]. The most damaging effects of climate change are floods, salinity intrusion, and droughts that are found to drastically affect crop productivity. Over the last

decades, climate changes with continuous and prolonged droughts, extreme temperature, and delayed rainfall are major causes affecting soil and water salinity. It is estimated that up to 20% of cultivated areas are affected worldwide by salinity [3].

Grapevine (*Vitis vinifera* L.) is one of the most important fruit species of the Vitaceae family, used mainly for wine, table grape, and raisin production. In most cases, it is growing grafted onto different rootstocks derived from North American *Vitis* species, mainly in regions where phylloxera is present. The practice of grafting was first used in grapevine because of phylloxera, a root-feeding aphid, that lives on vine roots and can totally damage vine plants of *Vitis vinifera* L. species. However, besides phylloxera resistances, the use of rootstocks also affects the interactions between vines and the environment in terms of nutrients uptake, salinity tolerance, and water stress resistance. Salinity in soils and irrigation water consists of an environmental perturbation that negatively affects grapevine growth and yield. High salinity causes severe problems in water uptake and availability of nutrients, increasing toxic-ion concentration in soils and vine plant tissues [4]. Furthermore, inhibition of grapevine growth and $CO_2$ assimilation by salinity have been related to changes in some physiology parameters as leaf stomatal conductance, electron transport rate, leaf water potential, and chlorophyll fluorescence [5,6]. The photosynthesis inhibition caused by the reduction of stomatal conductance as a direct salinity effect was reported [7].

Grapevine is moderately sensitive to high soil salinity, and damages caused by this stress are primarily related to chloride ions [8,9]. Grapevine rootstocks play an essential role in mineral nutrition, growth, and the yield of the vine. Arbabzadeh and Dutt [10] reported that the reduction in growth due to salinity seems to be related to the sum of the total cations in the leaves. Moreover, grapevine rootstocks exhibit differential degrees of tolerance in response to salinity and other abiotic stress [11]. Plants grown under saline conditions show an increase in Na and Cl concentrations, and a decrease in K, Ca, and Mg [12]. High Cl concentrations in plant tissues may negatively affect gas exchanges performed by plants since they often induce the occurrence of considerable effects on the net photosynthesis and stomatal conductance rates [5]. Some of the rootstocks have been rated as tolerant to salinity due to their ability to prevent Na and/or Cl uptake and translocation to scion [13]. Rootstocks obtained from different American *Vitis* species differ widely in their ability to exclude Cl and consequently in their capability to higher tolerant salinity. Fisarakis et al. [13], demonstrated that rootstocks derived from *Vitis Berlandieri* species had a great ability for Cl and/or Na exclusion, although this ability is reduced in *Berlandieri–Vinifera* hybrid rootstock (41B). Others reported an inverse relation between Cl and Na, suggesting that rootstocks transporting Cl readily should restrict Na transport and vice versa [14]. It has been reported that own-rooted Sultana vines irrigated with saline water suffered a significant yield reduction, whereas Sultana grafted to Ramsey and 1103 Paulsen have not shown any yield reduction [15].

Concerning the Greek grape growing, most of the vineyards are mainly located on the slope of hills and mountains or in areas described as semiarid, especially those located in Greek islands. There are several small islands where vines are grown under hot and drought Mediterranean climate. Summers are dry and hot while winters have moderate temperatures. Therefore, irrigation is crucial for quality fruit production [16]. Poor irrigation management can result in water stress, leading to unbalanced growth, reduced yields, and inferior fruit quality [17]. However, vine irrigation with low-quality groundwater and increased salt content, in combination with the use of drip irrigation and the increased summer temperature and/or poor soil drainage, may result in salt accumulation and salinity stress.

Since there are many contradictory data in the literature on the response of the different rootstocks to soil salinity, the main aim of this paper is to assess the effect of rootstock and scion variety on some physiological parameters and nutrient uptake under salinity. In a hydroponic vine growing experiment, two grapevine cultivars grafted onto two different rootstocks were assessed. It is important to mention that the two vine rootstocks used in our experiment, 1103 P and 101-14 Mgt, have been chosen because of their big differences in

terms of genetic characters. Regarding the vine varieties, Cabernet Franc and Merlot have been selected since they consist of two of the most common varieties cultivated worldwide.

## 2. Materials and Methods

### 2.1. Plant Material and Experimental Conditions

This experiment was conducted at the experimental farm of the Aristotle University of Thessaloniki, which is located 15 km southeast of Thessaloniki (Northern Greece, geographical coordinates: N40°32.267′ E22°59.885′). At the end of March 2017, one-year-old vine plantings of Merlot and Cabernet Franc cultivars grafted onto 1103 Paulsen and 101-14 Mgt rootstocks, or own-rooted, were planted in 2.5 L nursery bags containing a growing medium consisting of 30% low-phosphate sandy loam soil {5 mg kg$^{-1}$ P, low organic matter (1.2%), pH 7.0 and low total CaCO$_3$ (2.5%)}, 40% sable, and 30% turf. These nursery bags had been kept in a glasshouse for one month before being transferred outdoors. During the vegetative period, all vine plants in plastic bags received the usual cultivation practices. At the beginning of the following growing season, uniform plants were selected and transplanted in 10 L pots containing 9 L inert sand: perlite (1:1) medium. All vine plants were pruned to single shoots with two buds and the pots were placed outdoors. Six uniform vines with two developed shoots were selected for each scion–rootstock combination or own-rooted vines, subsequently transported in the experimental place and irrigated automatically every two days using drip irrigation system with 650 mL per plant, of modified 50% Hoagland No 2-nutrient solution [18]. Vine plants in pots were supported on a stake placed into the growing medium. The experiment was designed as a completely randomized 3 × 2 × 2 factorial, with two levels of salinity treatment (Control and 100 mM NaCl salinity), two vine cultivars, and two vine rootstocks or own-rooted vines with six replications.

Over the summer months, a green polyethylene external shade netting system was fitted and subsequently fixed with clips in order to prevent leaf and root temperature rising to very high levels. During physiology parameters measurements, the shade system could be easily removed, so as to allow maximum light exposure of plants.

The two vine rootstocks used in our experiment have been chosen according to the genetic character: the 1103 P rootstock, is a *V. berlandieri/V. rupestris* hybrid adapted to deep, well-drained calcareous soils, providing excellent protection against phylloxera. Moreover, the 101-14 Mgt rootstock, is a *V. riparia/V. rupestris* hybrid, tends to be easy to root and graft, providing also excellent protection against phylloxera, whereas it is not well adapted in calcareous soils.

NaCl was chosen for the salinity treatment because of the strong toxic effects of Cl ion when it is accompanied with the Na [19]. After the initial establishment, salinity treatment was applied for a period of 60 days (from 1 July to 30 August) by application of 650 mL 100 mM NaCl solution, adjusted to 10 dS m$^{-1}$ using an EC meter (HANNA HI 8733), three times a week. The electric conductivity of Irrigation water for control vines was 0.3 dS m$^{-1}$. Physiological parameters and leaf Chlorophyll content were measured in three different stages after the beginning of the salinity application (1 July, 1 August and 1 September).

### 2.2. Chlorophyll Content

For non-destructive chlorophyll estimation, a chlorophyll content meter (CM) CCM–200 (Opti-Sciences, Tyngsboro, MA, USA) was used. On each plot and sampling time, CM readings were taken in the different experimental plots on three fully expanded leaves, located on the basal nodes of the shoots. The average CCM-200 readings for each leaf were recorded. Immediately following CM-200 readings, the leaves were cut from the plant, sealed in plastic bags, and transported to the laboratory in a cooler for Chl determination. The total Chl (a + b) concentration in leaves was recorded according to Wintermans and De Mots [20]. Leaf discs of 0.5 g were extracted in 15 mL of ethanol (96%) and then placed in a water bath at 79.8 °C until complete discoloration, after about 2 h. The absorbance of the extract for chlorophylls a and b was measured at 665 and 649 nm, respectively, with

a spectrophotometer (Jenway Ltd., Essex, UK). Total chlorophyll for fresh material was determined according to the equation:

$$Cl\ (a + b)\ mg.\ g^{-1}\ FW = (6.10 \times A665 + 20.04 \times A649) \times 15/100FW$$

### 2.3. Water Status and Photosynthetic Activity

Vine water status was estimated by measurements of stem water potential (Ws) at three stage of the experimentation using a pressure chamber, according to Chone et al. [21]. For each measurement, a single leaf per plant was tightly zipped in a plastic bag to eliminate transpiration, with care taken to not clamp or damage the petiole. Aluminum foil was then placed around the bag, deflecting light and heat for at least 90 min before measurement, to allow equilibration of Ws. The PSII Chlorophyll fluorescence was measured in attached, 30 min dark-adapted leaves with a portable Chlorophyll fluorometer (PEA Hansatech Instruments Ltd., King's Lynn, UK) at midday. The light emitted from the LED is filtered using a near infrared radiation (NIR) filter to block any infra-red content, which could be seen by the detector. The single LED provides up to 3.500 $\mu$mols m$^{-2}$ s$^{-1}$ PPFD (Photosynthetic Photon Flux Density) with a peak wavelength of 627 nm. Minimum ($F_0$), Maximum (Fm), and variable (Fv) fluorescence yield parameters were automatically measured, and maximum quantum yield (Fv/Fm) was recorded.

Net assimilation rate (A) and stomatal conductance ($g_s$) were recorded from 11 a.m. to 1 p.m. simultaneously with Ws measurements using the LCi portable gas exchange system (ADC BioScientific Ltd., Hoddesdon, UK). Measurements were taken on four fully expanded sun-lit leaves per plot (Photosynthetic photon flux density > 1200 mol m$^{-2}$ s$^{-1}$, adjacent to those used for Ws).

### 2.4. Nutrient Element Composition and Growth Parameters

At the end of the experimental period, leaves, shoots, roots, and trunks were separated and all the plant tissues were subsequently washed and dried at 70 °C. Afterward, they were ground to a fine powder, to pass a 30-mesh screen. The dry weight of roots, shoots, and trunks (g) was also recorded at the end of the experiment. A portion of 0.5 g of the fine powder of each sample was incinerated in a muffle furnace at 515 °C for 5 h. Then, the ash was dissolved with 3 mL of 6 N HCl and diluted with double distilled water up to 50 mL. The concentrations of P, K, Ca, Mg, Na, Fe, Mn, and Zn were determined by ICP (Perkin Elmer-Optical Emission Spectrometer, OPTIMA 2100 DV) [22], while those of N, B, and Cl were determined by the Kjeldahl method [23] as well as by the methods of azomethine-H and Mohr [24,25], respectively. Macronutrient (N, P, K, Ca and Mg) and Na, Cl concentrations were expressed in % D.W., while those of micronutrients (Fe, Mn, Zn, and B) were expressed in mg kg$^{-1}$ D.W.

### 2.5. Total Phenolics

At the end of the salinity treatment period (60 days), leaf total phenolics were determined. Total phenolic (TP) concentration in leaf extracts was determined spectrophotometrically according to the Folin–Ciocalteu colorimetric method [26] calibrating against catechin standards and expressing in mg·g$^{-1}$ catechin equivalents (CE), of leaf fresh weight (f.w.). Analyses were performed in triplicate on each extract.

### 2.6. Statistical Analysis

The measured parameters were analyzed according to a randomized, complete block design. Data were analyzed by the software SPSS Version 24. The least significant difference test was used to detect differences between the means of the fixed effects at $p < 0.05$.

## 3. Results

### 3.1. Nutrient Concentrations in Plant Tissues

Sixty days after the beginning of NaCl salinity treatment, Cl and Na concentrations were higher in all parts of the vines (Table 1). Among the different parts, leaves showed the highest Cl concentrations (0.69–2.67% d.w.), whereas the roots showed the highest Na concentrations (0.046–1.87% d.w.). However, the rate of increase was drastically higher in roots than in leaves. According to the results, higher Na concentration was recorded in Cabernet Franc roots than in Merlot as well as in vines grafted onto 101-14 Mgt rootstock. In addition, salt treatment for 60 days duration increased Cl contents from 1.5 to 2.4-fold in leaves, compared to control. Increased leaf Cl concentrations were recorded in Cabernet Franc cultivar compared to Merlot. Regarding rootstock effect, a significantly increased Cl concentration was recorded in Merlot cultivar when grafted onto 101-14 Mgt. It was observed also that Na and Cl concentration in own-rooted vines was consistently higher compared with grafted ones. In addition, low Na and Cl concentration was observed in shoot and trunk parts of the vines.

**Table 1.** The effect of NaCl salinity on Na, Cl, and K concentration (% d.w.) in different plant parts of Merlot and Cabernet Franc vines on own roots or grafted to 1103 P and 101-14 Mgt rootstocks.

| **Leaves** | | | | | | | |
| --- | --- | --- | --- | --- | --- | --- | --- |
| | | **Merlot** | | | **Cabernet Franc** | | |
| **Salinity** | | **Na** | **Cl** | **K** | **Na** | **Cl** | **K** |
| Control | Own roots | 0.071 | 0.52 | 2.14 | 0.08 | 0.60 | 2.21 |
| | 1103 P | 0.063 | 0.69 | 1.84 | 0.08 | 0.60 | 2.51 |
| | 101-14 Mgt | 0.055 | 0.70 | 1.94 | 0.06 | 0.70 | 2.43 |
| 100 mM NaCl | Own roots | 0.400 | 2.31 | 2.31 | 0.53 | 2.67 | 2.68 |
| | 1103 P | 0.210 | 1.25 | 1.51 | 0.35 | 1.61 | 2.11 |
| | 101-14 Mgt | 0.510 | 1.69 | 1.25 | 0.78 | 1.75 | 1.92 |
| | LSD ($p < 0.05$) | 0.044 | 0.183 | 0.183 | 0.044 | 0.183 | 0.183 |
| | F | 6.102 | 72.292 | 13.576 | 6.102 | 72.292 | 13.576 |
| **Roots** | | | | | | | |
| | | **Merlot** | | | **Cabernet Franc** | | |
| **Salinity** | | **Na** | **Cl** | **K** | **Na** | **Cl** | **K** |
| Control | Own roots | 0.120 | 0.47 | 0.56 | 0.17 | 0.35 | 0.58 |
| | 1103 P | 0.046 | 0.30 | 0.47 | 0.21 | 0.34 | 0.49 |
| | 101-14 Mgt | 0.160 | 0.33 | 0.32 | 0.55 | 0.75 | 0.10 |
| 100 mM NaCl | Own roots | 1.57 | 1.26 | 0.40 | 1.18 | 1.25 | 0.39 |
| | 1103 P | 1.12 | 0.94 | 0.33 | 1.51 | 1.21 | 0.31 |
| | 101-14 Mgt | 1.87 | 1.21 | 0.28 | 1.82 | 1.62 | 0.30 |
| | LSD ($p < 0.05$) | 0.031 | 0.101 | 0.011 | 0.031 | 0.101 | 0.011 |
| | F | 26.346 | 20.909 | 52.943 | 26.346 | 20.909 | 52.943 |
| **Shoots** | | | | | | | |
| | | **Merlot** | | | **Cabernet Franc** | | |
| **Salinity** | | **Na** | **Cl** | **K** | **Na** | **Cl** | **K** |
| Control | Own roots | 0.04 | 0.09 | 0.60 | 0.04 | 0.08 | 0.52 |
| | 1103 P | 0.022 | 0.11 | 0.61 | 0.02 | 0.19 | 0.63 |
| | 101-14 Mgt | 0.015 | 0.10 | 0.60 | 0.03 | 0.14 | 0.44 |
| 100 mM NaCl | Own roots | 0.18 | 0.11 | 0.42 | 0.21 | 0.15 | 0.40 |
| | 1103 P | 0.11 | 0.18 | 0.56 | 0.71 | 0.29 | 0.46 |
| | 101-14 Mgt | 0.16 | 0.16 | 0.58 | 0.81 | 0.23 | 0.39 |
| | LSD ($p < 0.05$) | 0.028 | 0.026 | 0.038 | 0.028 | 0.026 | 0.038 |
| | F | 27.934 | 40.870 | 24.697 | 27.934 | 40.870 | 24.697 |

**Table 1.** *Cont.*

| Trunks | | | | | | | |
| --- | --- | --- | --- | --- | --- | --- | --- |
| | | Merlot | | | Cabernet Franc | | |
| **Salinity** | | **Na** | **Cl** | **K** | **Na** | **Cl** | **K** |
| Control | Own roots | 0.02 | 0.11 | 0.41 | 0.042 | 1.60 | 0.43 |
| | 1103 P | 0.032 | 0.38 | 0.17 | 0.036 | 0.75 | 0.24 |
| | 101-14 Mgt | 0.050 | 0.31 | 0.26 | 0.055 | 1.20 | 0.28 |
| 100 mM NaCl | Own roots | 0.52 | 0.23 | 0.33 | 0.085 | 0.81 | 0.31 |
| | 1103 P | 0.61 | 0.59 | 0.24 | 0.153 | 1.21 | 0.23 |
| | 101-14 Mgt | 0.75 | 0.67 | 0.21 | 0.158 | 0.92 | 0.23 |
| | LSD ($p < 0.05$) | 0.012 | 0.134 | 0.020 | 0.012 | 0.134 | 0.02 |
| | F | 70.668 | 20.041 | 19.841 | 70.668 | 20.041 | 19.841 |

Under the saline condition, the K concentration decreased in roots and leaves of the grafted vines whereas in own–rooted vines increased or remained unaffected. Among the different parts of the vines, leaves showed the highest K concentration, whereas the lowest concentration was recorded in trunk. Higher K values in leaves were recorded in own–rooted vines compared to grafted vines. Concerning the rootstock effect, it was found that vines raised on 1103 P rootstock under salinity, maintained higher (62%) K levels in leaves compared to 101-14 Mgt. In addition, according to our results, Cabernet Franc variety accumulated more K ions in leaves than Merlot. Salt treatments reduced Ca concentration from about 26% in leaves and roots to 15.27% in shoots, whereas the Ca content of the trunk was unaffected by salinity (Table 2). Among the different parts of the vines, leaves recorded a significantly higher accumulation of Ca. Calcium content in leaves was significantly affected by the stock-scion combinations in such a way that Cabernet Franc vines and 101-14 Mg rootstock showed higher Ca concentration.

**Table 2.** The effect of NaCl salinity stress on N, P, Ca, and Mg concentration (% d.w.) in different plant parts of Merlot and Cabernet Franc vines on own roots or grafted to 1103 P and 101-14 Mgt rootstocks.

| Leaves | | | | | | | | | |
| --- | --- | --- | --- | --- | --- | --- | --- | --- | --- |
| | | Merlot | | | | Cabernet Franc | | | |
| **Salinity** | | **N** | **P** | **Ca** | **Mg** | **N** | **P** | **Ca** | **Mg** |
| Control | Own roots | 2.19 | 0.24 | 2.14 | 0.39 | 2.07 | 0.33 | 2.65 | 0.41 |
| | 1103 P | 1.99 | 0.29 | 2.08 | 0.40 | 2.24 | 0.31 | 2.44 | 0.48 |
| | 101-14 Mgt | 2.02 | 0.21 | 2.43 | 0.46 | 1.99 | 0.36 | 2.51 | 0.54 |
| 100 mM NaCl | Own roots | 2.22 | 0.32 | 1.69 | 0.36 | 2.27 | 0.46 | 2.17 | 0.39 |
| | 1103 P | 1.65 | 0.41 | 1.29 | 0.35 | 1.91 | 0.38 | 1.79 | 0.41 |
| | 101-14 Mgt | 1.87 | 0.48 | 1.76 | 0.37 | 1.78 | 0.45 | 1.98 | 0.26 |
| | LSD ($p < 0.05$) | 0.299 | 0.034 | 0.130 | 0.0147 | 0.299 | 0.034 | 0.130 | 0.0147 |
| | F | 8.023 | 10.552 | 28.838 | 15.498 | 8.023 | 10.552 | 28.838 | 15.498 |
| Roots | | | | | | | | | |
| | | Merlot | | | | Cabernet Franc | | | |
| **Salinity** | | **N** | **P** | **Ca** | **Mg** | **N** | **P** | **Ca** | **Mg** |
| Control | Own roots | 0.58 | 0.27 | 0.91 | 0.27 | 0.66 | 0.31 | 1.12 | 0.32 |
| | 1103 P | 0.60 | 0.22 | 1.14 | 0.18 | 0.62 | 0.29 | 0.95 | 0.24 |
| | 101-14 Mgt | 0.78 | 0.31 | 1.11 | 0.23 | 0.70 | 0.28 | 1.31 | 0.26 |
| 100 mM NaCl | Own roots | 0.54 | 0.29 | 0.94 | 0.26 | 0.60 | 0.32 | 0.95 | 0.36 |
| | 1103 P | 0.59 | 0.24 | 0.51 | 0.21 | 0.57 | 0.32 | 0.73 | 0.26 |
| | 101-14 Mgt | 0.61 | 0.33 | 1.02 | 0.25 | 0.63 | 0.31 | 1.01 | 0.25 |
| | LSD ($p < 0.05$) | 0.059 | ns | 0.046 | 0.043 | 0.059 | ns | 0.046 | 0.043 |
| | F | 30.920 | | 17.253 | 26.728 | 30.902 | | 17.253 | 26.728 |

**Table 2.** *Cont.*

| | | **Shoots** | | | | | | | |
|---|---|---|---|---|---|---|---|---|---|
| | | **Merlot** | | | | **Cabernet Franc** | | | |
| **Salinity** | | **N** | **P** | **Ca** | **Mg** | **N** | **P** | **Ca** | **Mg** |
| Control | Own roots | 0.75 | 0.25 | 0.74 | 0.16 | 0.81 | 0.24 | 0.72 | 0.17 |
| | 1103 P | 0.65 | 0.22 | 0.75 | 0.13 | 0.71 | 0.23 | 0.65 | 0.18 |
| | 101-14 Mgt | 0.79 | 0.30 | 0.82 | 0.18 | 0.79 | 0.21 | 0.64 | 0.15 |
| 100 mM NaCl | Own roots | 0.66 | 0.24 | 0.65 | 0.16 | 0.70 | 0.31 | 0.55 | 0.14 |
| | 1103 P | 0.53 | 0.27 | 0.72 | 0.14 | 0.65 | 0.32 | 0.72 | 0.13 |
| | 101-14 Mgt | 0.66 | 0.31 | 0.51 | 0.13 | 0.68 | 0.26 | 0.53 | 0.14 |
| | LSD ($p < 0.05$) | 0.032 | 0.06. | 0.098 | 0.011 | 0.032 | 0.069 | 0.098 | 0.011 |
| | F | 29.798 | 36.480 | 4.776 | 27.932 | 29.798 | 36.480 | 4.776 | 27.932 |
| | | **Trunks** | | | | | | | |
| | | **Merlot** | | | | **Cabernet Franc** | | | |
| **Salinity** | | **N** | **P** | **Ca** | **Mg** | **N** | **P** | **Ca** | **Mg** |
| Control | Own roots | 0.49 | 0.22 | 0.76 | 0.16 | 0.50 | 0.22 | 0.64 | 0.13 |
| | 1103 P | 0.51 | 0.24 | 0.89 | 0.11 | 0.51 | 0.18 | 0.65 | 0.13 |
| | 101-14 Mgt | 0.55 | 0.23 | 0.82 | 0.17 | 0.55 | 0.24 | 0.63 | 0.16 |
| 100 mM NaCl | Own roots | 0.47 | 0.26 | 0.70 | 0.13 | 0.49 | 0.21 | 0.55 | 0.12 |
| | 1103 P | 0.47 | 0.25 | 0.75 | 0.10 | 0.46 | 0.23 | 0.72 | 0.15 |
| | 101-14 Mgt | 0.49 | 0.26 | 0.71 | 0.11 | 0.44 | 0.25 | 0.67 | 0.12 |
| | LSD ($p < 0.05$) | 0.017 | ns | ns | ns | 0.017 | ns | ns | ns |
| | F | 5.125 | | | | 5.125 | | | |

According to the results, the effect of salinity on leaf, shoot, and roots Mg concentration was significant. In this study, leaf Mg contents decreased 20.45% in salinity plots. A similar decrease was observed in roots and shoots (about 14%), however, remained unaffected in trunks. A comparison between vines grafted to different rootstocks showed that increased leaf Mg concentration was recorded with the 101-14 Mgt rootstock.

Generally, P concentrations in different parts of the vines were increased in salinity treatment (Table 2). However, the rate of increase was drastically higher in leaves than in other parts. So, 66% and 29% increase of P concentration was observed in leaves of Merlot and Cabernet Franc, respectively. In addition, the same trend was also observed in the other parts of the vine as well. The effect of the rootstock on leaf P concentration was significant. It was found that vines grafted onto 101-14 Mgt rootstock had increased P concentrations.

Salinity treatment had a significant decreasing effect on leaf N contents of grafted vines, while it was unaffected on its own root ones. For this parameter, no significant differences were found between the tested cultivars and rootstocks. In the other parts of the vine, the percentage of tissue nitrogen also decreased with the salinity.

The findings related to Fe, Zn, Mn, and B micronutrient contents of leaves, roots, shoots, and trunk are shown in Table 3.

When compared to control plants, salinity caused significant decreases in Fe content in leaves, but an inverse tendency was observed in the other parts. A significant increase in roots was observed where no differences were found in shoots and trunks of the vines. Salinity increased the Zn content in all parts of the vines. As with Zn, Mn and B concentrations tended also to increase but some variation was observed in the different parts.

For Mn, no significant differences were found between control and salinity in leaves, whereas higher values were recorded in roots, shoots, and trunks in the salinized samples. Moreover, B concentration of the leaves in Merlot salinity plots increased 34% compared to control, whereas in Cabernet Franc it was increased 21%. In contrast, salinity decreased Boron concentration in roots, shoots, and trunks.

**Table 3.** The effect of NaCl salinity stress on Fe, Mn, Zn, and B concentration (mg kg$^{-1}$ d.w.) in different plant parts of Merlot and Cabernet Franc vines on own roots or grafted to 1103 P and 101-14 Mgt rootstocks.

| | | Leaves | | | | | | | |
|---|---|---|---|---|---|---|---|---|---|
| | | **Merlot** | | | | **Cabernet Franc** | | | |
| **Salinity** | | **Zn** | **Mn** | **B** | **Fe** | **Zn** | **Mn** | **B** | **Fe** |
| Control | Own roots | 16.24 | 86.85 | 44.32 | 105.07 | 13.03 | 72.02 | 40.30 | 107.95 |
| | 1103 P | 12.02 | 56.84 | 26.32 | 122.51 | 15.51 | 58.23 | 32.61 | 140.50 |
| | 101-14 Mgt | 12.32 | 66.86 | 40.24 | 126.69 | 11.46 | 42.40 | 38.34 | 82.91 |
| 100 mM NaCl | Own roots | 18.05 | 78.65 | 50.12 | 77.03 | 18.10 | 90.11 | 44.29 | 79.12 |
| | 1103 P | 15.29 | 52.45 | 48.16 | 82.68 | 17.86 | 49.28 | 40.61 | 109.84 |
| | 101-14 Mgt | 13.31 | 64.54 | 48.80 | 96.40 | 16.47 | 59.60 | 50.03 | 67.33 |
| | LSD ($p < 0.05$) | 2.47 | ns | 2.440 | 7.301 | 2.47 | ns | 2.440 | 7.301 |
| | F | 15.666 | | 8.990 | 23.351 | 15.666 | | 8.990 | 23.351 |
| | | **Roots** | | | | | | | |
| | | **Merlot** | | | | **Cabernet Franc** | | | |
| **Salinity** | | **Zn** | **Mn** | **B** | **Fe** | **Zn** | **Mn** | **B** | **Fe** |
| Control | Own roots | 13.40 | 12.75 | 13.02 | 138.15 | 18.91 | 26.99 | 14.41 | 423.13 |
| | 1103 P | 16.5 | 19.24 | 14.54 | 248.82 | 10.12 | 16.34 | 13.60 | 348.13 |
| | 101-14 Mgt | 12.07 | 19.80 | 19.17 | 67.46 | 12.07 | 19.80 | 19.74 | 297.03 |
| 100 mM NaCl | Own roots | 24.44 | 22.37 | 12.39 | 297.70 | 35.97 | 28.26 | 12.68 | 482.51 |
| | 1103 P | 16.08 | 23.42 | 13.90 | 288.60 | 18.11 | 24.66 | 14.94 | 444.80 |
| | 101-14 Mgt | 15.98 | 39.06 | 20.20 | 470.50 | 19.92 | 61.62 | 26.58 | 314.54 |
| | LSD ($p < 0.05$) | 2.316 | 3.893 | 1.481 | 38.992 | 2.316 | 3.893 | 1.481 | 38.992 |
| | F | 256.49 | 28.803 | 75.259 | 17.687 | 256.49 | 28.803 | 75.259 | 17.687 |
| | | **Shoots** | | | | | | | |
| | | **Merlot** | | | | **Cabernet Franc** | | | |
| **Salinity** | | **Zn** | **Mn** | **B** | **Fe** | **Zn** | **Mn** | **B** | **Fe** |
| Control | Own roots | 28.17 | 22.89 | 11.40 | 18.32 | 23.17 | 17.92 | 12.76 | 24.39 |
| | 1103 P | 20.17 | 16.03 | 11.50 | 39.85 | 20.17 | 17.59 | 13.40 | 38.07 |
| | 101-14 Mgt | 18.37 | 22.89 | 13.87 | 27.87 | 21.37 | 15.87 | 15.69 | 48.01 |
| 100 mM NaCl | Own roots | 31.22 | 23.87 | 9.66 | 24.98 | 27.22 | 25.65 | 9.92 | 34.52 |
| | 1103 P | 24.54 | 17.89 | 12.44 | 14.74 | 23.54 | 18.06 | 10.97 | 20.69 |
| | 101-14 Mgt | 24.97 | 23.87 | 13.99 | 18.52 | 26.97 | 20.93 | 11.55 | 32.85 |
| | LSD ($p < 0.05$) | 9.257 | 5.507 | 0.595 | ns | 9.257 | 5.95 | 0.595 | ns |
| | F | 22.217 | 12.090 | 28.389 | | 22.217 | 12.090 | 28.389 | |
| | | **Trunks** | | | | | | | |
| | | **Merlot** | | | | **Cabernet Franc** | | | |
| **Salinity** | | **Zn** | **Mn** | **B** | **Fe** | **Zn** | **Mn** | **B** | **Fe** |
| Control | Own roots | 22.4 | 18.87 | 12.88 | 142.4 | 20.61 | 22.59 | 11.40 | 100.73 |
| | 1103 P | 16.97 | 23.09 | 11.40 | 95.51 | 10.95 | 13.59 | 11.50 | 79.27 |
| | 101-14 Mgt | 12.55 | 24.78 | 13.71 | 160.93 | 10.72 | 24.87 | 13.87 | 52.31 |
| 100 mM NaCl | Own roots | 23.31 | 13.78 | 12.51 | 88.40 | 20.95 | 16.21 | 9.66 | 57.68 |
| | 1103 P | 17.62 | 22.46 | 10.09 | 181.1 | 15.53 | 14.94 | 12.44 | 112.23 |
| | 101-14 Mgt | 13.49 | 16.24 | 10.64 | 88.40 | 12.68 | 16.24 | 13.99 | 77.82 |
| | LSD ($p < 0.05$) | 2.134 | 2.850 | 0.729 | ns | 2.134 | 2.850 | 0.729 | ns |
| | F | 9.098 | 5.534 | 4.832 | | 9.098 | 5.534 | 4.832 | |

### 3.2. Chlorophyll Content

Leaf Chlorophyll content was measured on 1st, 30th, and 60th day of the experimentation using both analytical and non-destructive methods (CCM-200 index). Based on the results (Figure 1), salinity decreased both leaf Chlorophyll concentration and CCM-200 Chlorophyll index, especially after 30 and 60 days of the beginning of experimentation. Leaf total chlorophyll concentration in vines under salinity decreased from 42% on the 30th day to 40% on the 60th day (Figure 1A). At the above two stages, chlorophyll reduction was significantly higher in 101-14 Mgt rootstock when compared to 1103 P. Sixty days after the beginning of salinity treatment, chlorophyll concentrations were least in the case of 101-14 Mgt rootstock, whereas higher values were recorded in case of 1103 P. Like chlorophyll concentration, CCM-200 index affected by salinity (Figure 1B). A similar trend for the relative chlorophyll content was observed in salinity plots.

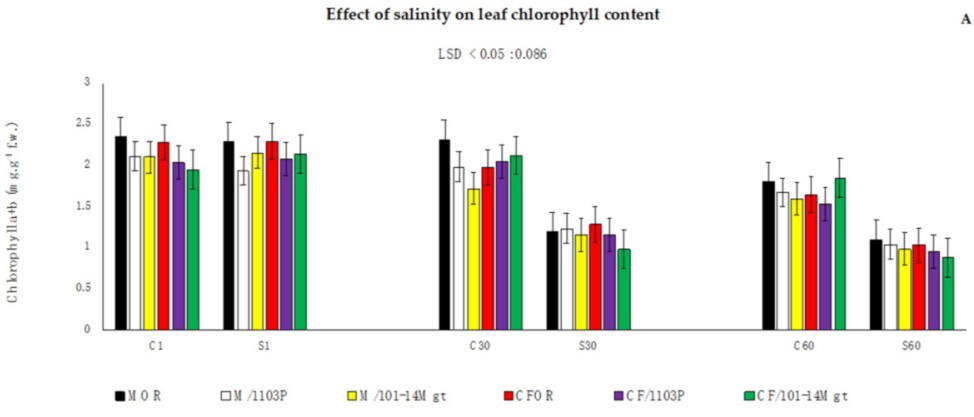

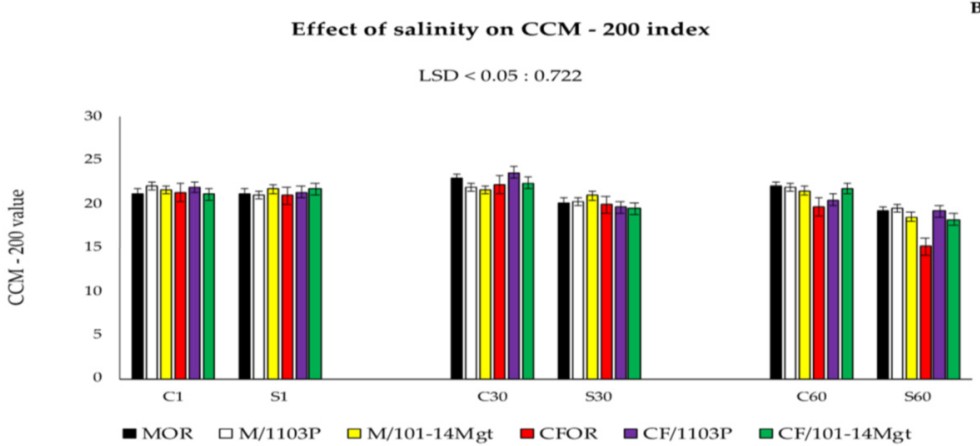

**Figure 1.** Leaf chlorophyll content (**A**) and CCM-200 Chlorophyll index (**B**) at 1st, 30th, and 60th day after the beginning of NaCl salinity treatment in own-rooted (OR) or grafted to 1103 P and 101-14 Mgt rootstocks of Merlot (M) and Cabernet Franc (CF) vine varieties. C: control, S: salinity. 1, 30, 60 (at day one, thirty and sixty respectively) LSD: Least Significant Differences. Chl a+b (F:34.152). CCM-200 (F: 9.076). Vertical bars indicate standard errors.

### 3.3. Total Phenolics

The leaf total phenolic contents in the examined samples ranged from 26.41 to 60.33 mg g$^{-1}$ (Figure 2). With salinity treatments, the accumulation of these compounds was significantly increased 59.28% for Merlot vines and 69.92% for Cabernet Franc. It is important to mention that under salinity treatment, Merlot on own roots or grafted onto

1103 P, contained the highest phenolic concentration in leaves, whereas Cabernet Franc on own roots or grafted to 101-14 Mgt the lowest.

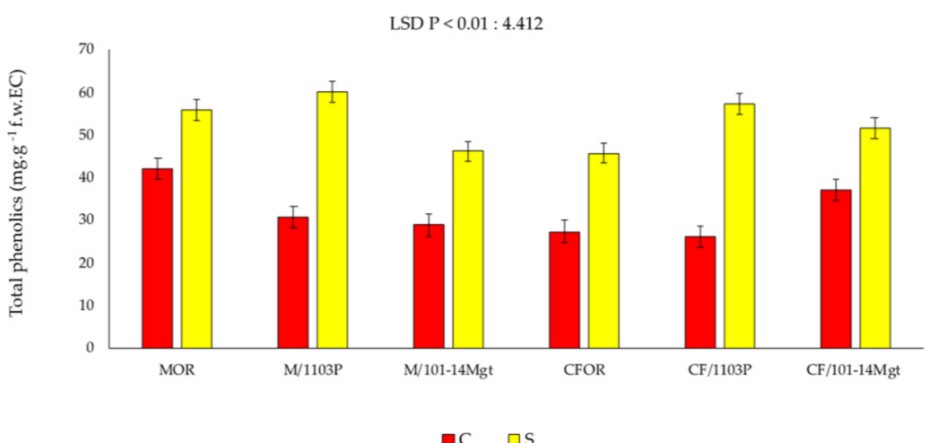

**Figure 2.** Leaf total phenolic content after sixty days of NaCl salinity treatment in own-rooted (OR) or grafted to 1103 P and 101-14 Mgt rootstocks of Merlot (M) and Cabernet Franc (CF) vine varieties. C: control, S: salinity, LSD: Least Significant Differences, (F: 22.717). Vertical bars indicate standard errors.

*3.4. Water Status and Photosynthetic Activity*

Steam water potential decreased gradually during the period of the experimentation, reaching minimum values after sixty days of salinity treatment (Figure 3). The analysis of variance showed a significant effect of the experimentation day, rootstock, and salinity treatment. Salinity significantly decreased steam water potential in all cases of grafted or own-rooted vines. However, comparing the two rootstocks, 101-14 Mgt rated always lower values of steam water potential.

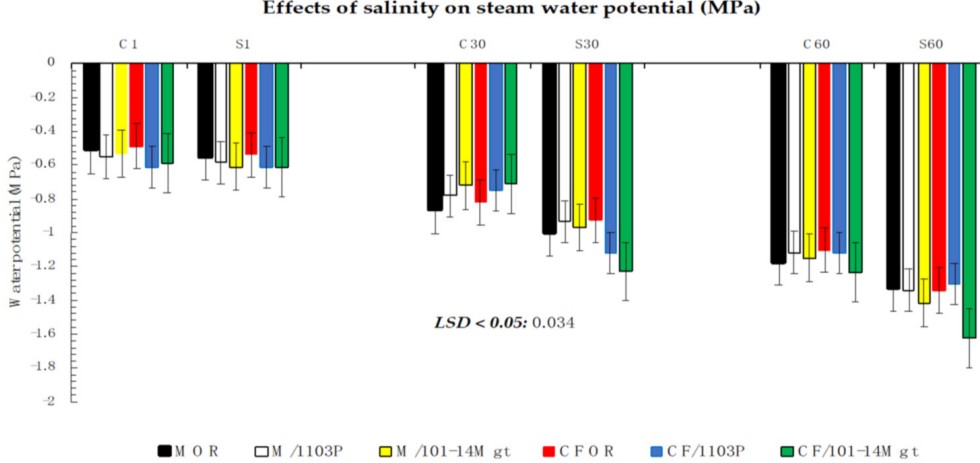

**Figure 3.** Steam water potential at 1st, 30th, and 60th day after the beginning of NaCl salinity treatment in own-rooted (OR) or grafted to 1103 P and 101-14 Mgt rootstocks of Merlot (M) and Cabernet Franc (CF) vine cultivars. C: control, S: salinity, 1, 30, 60 (day one, thirty and sixty, respectively) LSD: Least Significant Differences, (F: 145.983). Vertical bars indicate standard errors.

Photosynthetic activity was evaluated by measuring net assimilation rate, stomatal conductance, and PSII Chlorophyll fluorescence (ChF). According to our results, photosynthetic activity is considerably reduced by salinity treatment but only at thirty and sixty

days after treatment. No significant differences were observed at the beginning of the experimentation (data not shown). Net assimilation rate was generally lower in salinity vines ranging between 5.5 and 6.73 μmol $CO_2$ $m^{-2}$ $s^{-1}$ at day thirty, and 2.63 and 4.10 at day sixty (Figure 4A). Analysis of variance revealed that the main effect of scion variety, rootstock, and salinity as well as their interaction were significant ($p < 0.001$). The adverse effect on photosynthetic activity was associated with a significant decrease in stomatal conductance (Figure 4B). Among the two tested rootstocks and scion variety, 101-14 Mgt and Cabernet Franc showed the lowest values, respectively. As shown in Table 4, the changes in chlorophyll fluorescence parameters were relatively stable in the control vines compared to the salinity ones. On the contrary, from day thirty to day sixty, salt-treated vines had a decreased PSII maximum quantum yield (Fv/Fm). However, it was only towards the end of the experimentation that net changes were found. Sixty days after the beginning of salt treatments, Fv/Fm ratios were least in case of own-rooted Cabernet Franc variety or grafted to 101-14 Mgt rootstock.

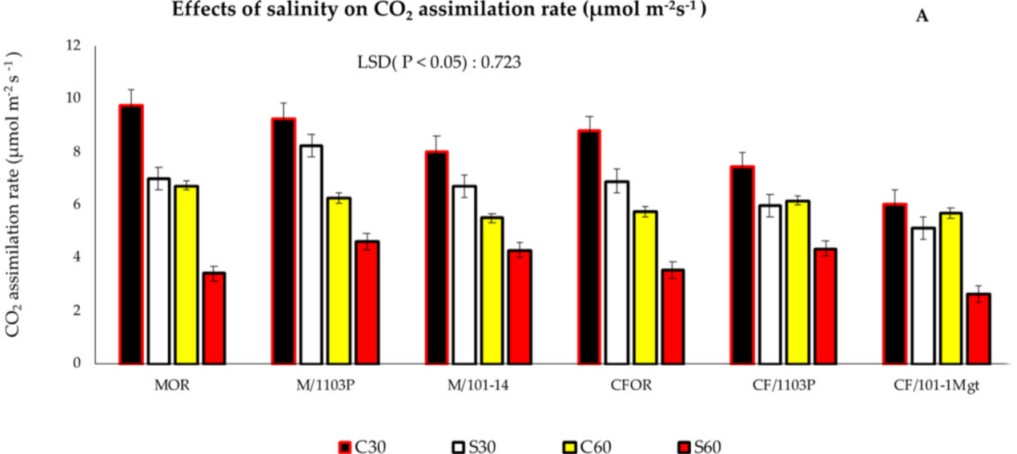

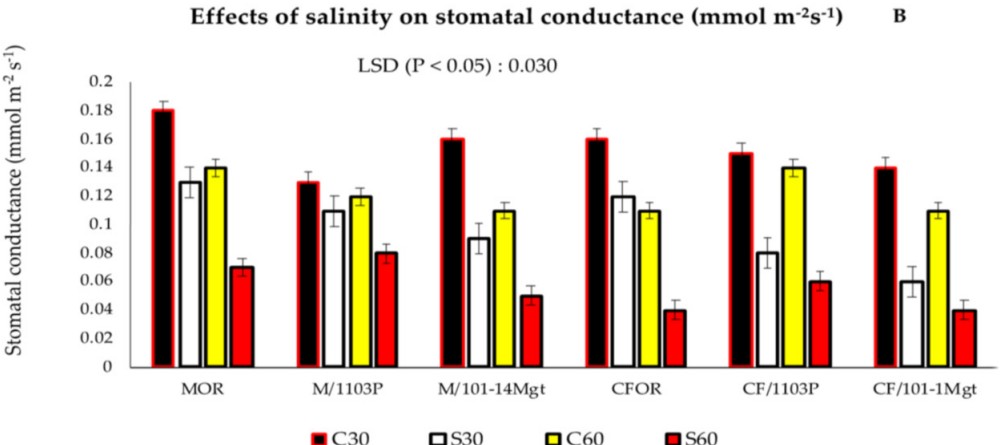

**Figure 4.** Effects of salinity on net assimilation rate (**A**) and stomatal conductance (**B**) at 30th and 60th day after the beginning of NaCl salinity treatment in own-rooted (OR) or grafted to 1103 P and 101-14 Mgt rootstocks of Merlot (M) and Cabernet Franc (CF) vine varieties C: control, S: salinity C30, S30 C60, S60 (day thirty and sixty, respectively) LSD: Least Significant Differences, A (F: 21.391), B (F:1.008). Vertical bars indicate standard errors.

**Table 4.** Chlorophyll fluorescence after thirty and sixty days of NaCl salinity treatment on own-rooted or grafted to 1103 P and 101-14 Mgt rootstocks of Merlot and Cabernet Franc vine varieties.

| | | Merlot | | Cabernet Franc | |
|---|---|---|---|---|---|
| Salinity | | Thirty Day | Sixty Day | Thirty Day | Sixty Day |
| Control | Own roots | 0.811 | 0.711 | 0.806 | 0.737 |
| | 1103 P | 0.818 | 0.759 | 0.812 | 0.737 |
| | 101-14 Mgt | 0.821 | 0.772 | 0.819 | 0.749 |
| 100 mM NaCl | Own roots | 0.795 | 0.561 | 0.801 | 0.598 |
| | 1103 P | 0.819 | 0.702 | 0.802 | 0.715 |
| | 101-14 Mgt | 0.815 | 0.724 | 0.811 | 0.653 |
| | | LSD $p < 0.05$: 0.059, F: 12.947 | | | |

### 3.5. Growth Parameters

Analysis of variance revealed no significant effect of salinity on growth parameters, whereas the effects of rootstock and scion cultivar had a significant effect on shoot, root, and trunk dray weight ($p < 0.001$). Differences between rootstocks, and between scion cultivars, as well as the rootstock x scion cultivar interaction, were significant. Increased plant vigor showed Merlot cultivar and vines grafted to 1103 P rootstock (Table 5).

**Table 5.** The effect of Na Cl salinity stress on shoot, trunk, and root dry weight (g) of own-rooted or grafted to 1103 P and 101-14 Mgt rootstocks of Merlot and Cabernet Franc vine cultivars.

| | | Merlot | | | Cabernet Franc | | |
|---|---|---|---|---|---|---|---|
| Salinity | | Shoots | Trunks | Roots | Shoots | Trunks | Roots |
| Control | Own roots | 48.48 | 62.33 | 60.75 | 35.28 | 54.61 | 57.27 |
| | 1103 P | 55.98 | 59.09 | 94.82 | 42.42 | 71.53 | 90.48 |
| | 101-14 Mgt | 47.03 | 48.11 | 75.28 | 38.54 | 62.35 | 70.41 |
| 100 mM NaCl | Own roots | 47.31 | 59.61 | 60.25 | 33.24 | 57.16 | 57.82 |
| | 1103 P | 52.17 | 57.86 | 91.92 | 40.43 | 72.36 | 89.41 |
| | 101-14 Mgt | 46.26 | 46.50 | 74.95 | 41.10 | 60.94 | 70.82 |
| | LSD ($p < 0.05$) | 4.75 | 6.78 | 13.842 | 4.75 | 6.78 | 13.842 |
| | F | 13.196 | 11.665 | 13.266 | 13.196 | 11.665 | 13.266 |

## 4. Discussion

Salinity treatment affected all major physiological processes related to vine growth such as water status, photosynthesis, ion uptake, and transport. Among them, ion uptake affects plant growth in different ways, which can lead to deficiency or toxicity phenomena [4,27,28]. This nutritional aspect for salinity effects may be related to the availability, competition in absorption, transmission, or distribution of nutrients in the plant.

### 4.1. Nutrient Concentrations in Plant Tissues and Vine Growth

Table 1 summarizes Na, Cl, and K concentrations in the different parts of the vines. Salinity stress caused an increase in levels of Na and Cl in all parts of the vines. The highest Cl concentration was found in the leaves followed by the roots, whereas the highest Na concentrations were found in roots. It has been reported that high uptake and root-to-shoot transport of chloride resulted in its excessive accumulation in leaves, causing impaired leaf function and damage under salinity [7,29]. Cl and Na exclusion by roots may prevent their accumulation in leaves, thus contributing to salt tolerance [30]. Additionally, a decrease of K, Ca and Mg was detected in salinity treatments. In our experiment, salt injury symptoms (Figure 5), appeared first on own-rooted Cabernet Franc variety at about 40 days after the

salinity treatment followed by Cabernet Franc grafted to 101-14 Mgt and Merlot. In control vines, leaf Na concentrations were very low (<0.1%) in both vine varieties and below toxic threshold interval of 0.25–0.50% reported by Stevens et al. [31] and walker et al. [29], whereas in salinity were consistently higher. In contrast, leaf Cl concentrations were much higher and tended to increase with salinity, being always within the toxic interval for vine. According to Ehlig [32], levels above 1.2% are toxic for the vine.

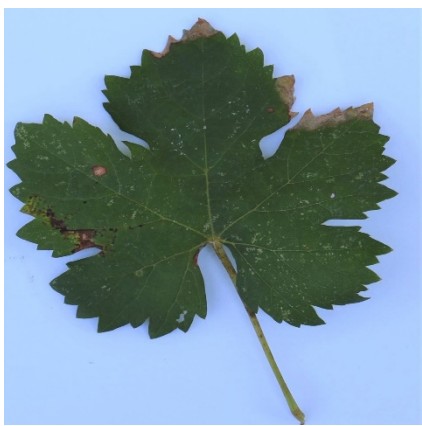

**Figure 5.** First leaf necrosis symptoms in own-rooted Cabernet Franc vines after 40 days of salt treatment.

As we can see in Table 1, own-rooted vines accumulated higher values of Cl ions in leaves compared to grafted ones. Likewise, higher values were recorded in Cabernet Franc as well as in vines grafted onto 101-14 Mgt rootstock. It has been reported by several researchers that the rootstock tolerance in salinity due to their ability to prevent Na and/or Cl uptake, and translocation to aerial parts of the vines [33]. According to recent studies, a valuable list of candidate genes mediates shoot Cl exclusion [34]. It has also been mapped a quantitative trait locus (QTL) that controls Na exclusion in rootstocks and the causal gene (VisHTK1;1) underlying that trait [35].

So, 101-14 Mgt can be considered as a more sensitive rootstock compared to 1103 P. Moreover, Cl concentration in the leaves of Merlot was lower than that in Cabernet Franc. Based on this difference and the above-mentioned visual leaf burn toxicity symptoms, we can suggest that Merlot should have a higher tolerance to salinity than Cabernet Franc. Among the assessed rootstocks, it seems that 1103 P was the best excluder for Na and Cl concentrations in leaves.

Except for higher Cl and Na leaf concentrations, sensitive rootstocks accumulate more total cations [10]. According to the results of the present study, 101-14 Mgt rootstock accumulated more cation in leaves compared to 1103 P (Table 1). Potassium is an essential factor in protein synthesis, glycolytic enzymes, and photosynthesis [36]. Because salinity affects plant growth similarly through water deficit, K is equally important for maintaining the turgor pressure in plants under salinity and drought stress. It has been reported that higher Na and Cl uptake competes with the uptake of other nutrient elements, especially K, and decreases K, Ca, and Mg levels in several plant species [37–39]. According to Samra [40], lower sodium/potassium ratios in the vines indicate tolerance to salinity stress. Additionally, higher K/Na ratios will also improve the resistance of the plant to salinity [41]. In the present study, higher K/Na ratios were recorded with 1103 P. Upadhyay et al. [42], reported that sodium/potassium ratios in leaf blade and petiole were least in case of 110 R and 1103 P rootstocks, whereas higher values were recorded in the case of other sensitive rootstocks. Gong et al. [43] reported that more than one gene is related to K and Na accumulation in different biotypes. Moreover, the capability to compartmentalize Na ions in the vacuole is also related to saline tolerance of some biotypes. A tonoplast-localized $Na/H^+$ exchanger 1 (NHX1) plays an important role in the vacuolar sequestration of Na ions [44].

The result of the present study showed that salinity treatment caused an increase in Na concentration, and a decrease in K and Ca concentration in all cases. However, the increase in Na concentration in the sensitive 101-14 Mgt rootstock was higher than 1103 P, but the decrease in K concentration in 101-14 Mgt was higher than in tolerant 1103 P (35% and 18%, respectively). These findings agree with other reports concerning other plant species [45]. Calcium is important during salt stress in cell wall and membranes and regulates the growth and development of plants [46]. Salt treatments reduced Ca concentration in leaves and roots, but higher levels were recorded in leaves and roots of the more sensitive rootstock (101-14 Mgt) and Cabernet Franc scion variety (Table 2). Experimental evidence implicates Ca function in salt adaptation. Externally supplied Ca reduces the toxic effects of salinity presumably by facilitating higher K/Na selectivity [47].

Other important nutrients like Mg and P were significantly affected by salinity but in a different way (Table 2). Leaf Mg accumulation decreased in stressed vines whereas P accumulation increased. The role of Mg in chlorophyll structure and as an enzyme co-factor is considerable. Decreases of Mg content of leaves have been reported upon salt accumulation, suggesting a decreased chlorophyll levels [48]. Phosphorus concentration in leaves and shoots increased by salinity while in roots and trunks remained unaffected (Table 2). These findings agree with those reported by Downton [49], Stevens et al. [50], and Sivritepe et al. [51]. Different researchers report contradictory results concerning differences in plant P content in salt stress conditions. Some studies in other plant species indicated that salinity, either increased the P content in plant tissue [52] or had no effect [53], nevertheless, other studies indicated a decrease [54]. The contradictory findings could be attributed to different experimental conditions or even to transport problems [55]. Phosphorus is transported primarily in the phloem [56,57] and different species seem to control the transfer of P from roots to shoots differently. Sixty days after salinity treatments, N concentration decreased in all parts of the grafted vines, while it was unaffected on own-rooted ones. Numerous studies have shown that excess salts can reduce the accumulation of nitrogen in plants [11,53,58,59]. There are authors who have attributed this reduction to the antagonism between Cl and $NO_3^-$ [60] and those who explain it by reduced water uptake [61]. However other scientists reported less effect or even the opposite [49,51,62]. These contradictory findings could be ascribed to different experimental conditions or to different sampling techniques. Moreover, it is well-known that the results of salinity conducted in hydroponic or sand cultures, as in the present experiment, contrast markedly with studies where plants were grown in soil.

The analysis of variance revealed a significant difference between the control and saline treatment for Fe, Zn, Mn, and B. Salinity caused a significant decrease in leaf Fe as well as a significant increase in roots (Table 3). In both rootstocks, the foliar Fe tended to decrease with salinity. The lowest leaf Fe concentration (67.33 mg kg$^{-1}$ d.w) occurred in Cabernet Franc variety when grafted to 101-14 Mgt rootstock. Regarding the shoot and trunk tissues, there was no significant difference between the Fe concentrations in the control and salinity treatments. Among the different plant parts, roots showed the highest Fe concentrations.

The results of the present study showed that salinity caused an increase in Zn concentration in all parts. Increased B concentration was recorded in leaves of salinity plots and decreased concentration in roots, shoots, and trunks. Among the different plant parts, leaves showed the highest B concentrations. Concerning the B compartmentalization in different parts, the results agree with those reported by Nikolaou et al. [63]. Finally, Mn concentrations in saline-treated vines were higher in all parts of the vines compared to the control. It is difficult to suggest a mechanism to explain the salinity impacts on micronutrients because of a complex relationship between salinity and micronutrient uptake. According to Grattan and Grieve [64], many factors such as plant species, the kind of plant tissue, level of salinity stress and composition, micronutrient concentration in the growth medium, growth conditions, and stress duration could influence micronutrient uptake and transport.

Regarding the growth parameters, salinity had no effect on shoot, trunk, and root dry weight. This could be attributed to the sort time duration of salinity treatment (Table 5). As we will discuss further, salinity affected photosynthetic activity and stem water potential from thirty to sixty days after the beginning of experimentation. On contrary, the rootstocks and scion variety vary significantly with respect to the dry weight of roots and shoots. Merlot cultivar and vines grafted to 1103 P rootstock showed increased dray weights for the different plant parts.

### 4.2. Chlorophyll Pigments, Water Status, and Photosynthetic Activity

Salinity significantly decreased A and $g_s$ at thirty and sixty days after treatment (Figure 4). It is noteworthy that photosynthetic mechanisms are regulated by several external and internal factors such as, water potential and dehydration of cell membranes which reduce their permeability to $CO_2$, reduction of $CO_2$ supply because of hydro-active closure of stomata, enhanced senescence induced by salinity, salt toxicity, and photosynthetic pigment degradation [48,65]. Rootstocks exhibit different degrees of tolerance in response to salinity [66]. Vines receiving salt treatment accumulated increased values of Na and Cl into leaves over the course of the experiment, and necrosis was evident on the margins of some of the leaves by day 40 (Figure 5). Except for higher Cl and Na concentrations, sensitive rootstocks accumulate more total cations [10]. According to our results, 101-14 Mgt rootstock accumulated more cations in leaves compared to 1103 P. At the same time, a considerable drop of stem water potential was observed (Figure 3). Under high salinity, as in our experiment (EC:10 dS m$^{-1}$), vine roots are exposed to high osmotic stress which decreases plant water availability. Salinity decreased both leaf Chlorophyll concentration and CCM-200 Chlorophyll index especially after 30 and 60 days of the beginning of experimentation (Figure 1). In this study, the close association between CCM-200 meter and leaf Chl content proved that relative chlorophyll measurement could be used for rapid and cost-effective chlorophyll assessment [67]. An inhibitory effect of salinity on Chl synthesis or acceleration of Chl degradation was reported by Reddy and Vora [68]. Zhang et al. [69] reported that salinity induced swelling of chloroplast thylakoids and caused destructions of the chloroplast envelope. Biosynthesis of Chl is also inhibited when Na and/or Cl concentrate in the chloroplasts [70,71]. The symptoms of salt stress in grapevines include a reduction in stomatal conductance and photosynthesis, and leaf burn, which are generally related to an increase in leaf and shoots Cl rather than in Na concentration in plant tissues [29,72]. Moreover, reduced uptake of nitrate in combination with osmotic stress may explain the inhibitory effect of excess salts on photosynthesis. Chloride ion inhibits photosynthesis by inhibiting the uptake of nitrate nitrogen [13]. Abd El Baki et al. [73] and Flores et al. [74] reported that increased Cl ions under salt stress decrease Nitrate reductase activity in many plant species and consequently reduce $NO_3^-$ concentration in leaves. Stomatal conductance is also reduced under salinity and this consists of a way for plants to reduce water losses by leaf transpiration. The strong decrease in Ws observed from thirty to sixty days after the beginning of experimentation, was accompanied by a parallel decrease in stomatal conductance and photosynthetic rate (Figure 4). Moreover, severe reduction of photosynthesis at high salinity levels may attribute to the biochemical and photochemical capacities of the leaf (nonstomatal limitations) [75,76]. This may be due to an inhibition of several enzymes related to photosynthesis such as Rubisco, which has been reported to decrease under salt stress [77]. Net photosynthetic rate and the ratio of ribulose biphosphate carboxylase (Rubisco) activity to that of phosphor-enol pyruvate carboxylase (PEPC), decrease under salinity [78]. Decreased Rubisco regeneration could limit electron capture and transport from photosystem II (PSII) to electron acceptors [79]. PSII maximum quantum yield (Fv/Fm) measurements (Table 4) represent the differences in the efficiency of photosystem II in vine plants affected by salinity. Lower Fv/Fm was recorded in the salt-stressed compared to the control vines, but only towards the end of the experimentation period. After thirty days of salt treatment, values of Fv/Fm were relatively high for both control and salt stress treatments (0.795–0.819) in all cases, while at the

end of experimentation, differences among the control and treated vines were significant. It should be noted that very low values were recorded in Cabernet Franc variety when grafted to 101-14 Mgt rootstock (0.653). Generally, the stressed vines at this stage showed marginal leaf-borne symptoms. In our study, visible senescence and marginal leaf necrosis did not appear before forty days after the effect of salinity. However, according to our results, net assimilation rate and stomatal conductance decreased earlier than chlorophyll fluorescence. It has been reported by Nieva et al. [80] that the gas exchange process may be much more sensitive to salinity than photosystem II efficiency. Therefore, we hypothesized that these results could be due to the existence of other salt tolerance mechanisms. Based on the results of the present work, salinity treatments increased the accumulation of phenolic substances (Table 2). One of the most important aptitudes of these substances is the antioxidant activities [81]. Additionally, these compounds are accumulated to respond to the increases of reactive oxygen species (ROS) under salt stress [82–84]. Reactive oxygen species (ROS) are detrimental to cells at high concentrations because they cause oxidative damage to membrane lipids, proteins, and nucleic acids [85,86], resulting in inhibition of electron capture and transport from photosystem II (PSII). The results obtained from this research showed higher levels of the phenolic substances in salinity stressed vines. Moreover, in the Merlot variety, like in vines grafted to 1103 P, the leaf phenolic content was higher compared to Cabernet Franc grafted onto 101-14 Mgt rootstock. In addition to phenolic effects, K can increase resistance to several environmental stresses because it contributes to the detoxification of ROS [87]. The present study showed that the rootstock employed exerts a great influence on grapevine response to salinity. However, the selection of new rootstocks could be an interesting solution to overcome some of the risks associated with salinity, such as the excess of Cl and Na in leaves and bunches [88].

## 5. Conclusions

The results obtained from this research showed that 10 dS m$^{-1}$ salt applications affected Na and Cl leaf concentrations in such a way that leaf injury symptoms appeared at about 40 days after the treatments. At the end of the experimentation period, leaf Na and Cl were within the toxic intervals for the vine. Salinity and rootstock also affected the concentration and compartmentation of the most important nutrient elements. Chlorophyll content, steam water potential, and photosynthetic activity decreased in both cultivars. Scion cultivar and rootstock had a significant effect on the most measured parameters. Consequently, higher values of Chloride and Sodium concentrations were recorded in leaves of Cabernet Franc cultivar, like in vines grafted onto 101-14 Mgt rootstock, whereas earlier toxicity symptoms appeared. The results of this study showed that 101-14 Mgt rootstock and own-rooted vines could accumulate much faster Chloride ions in leaves. In fact, 1103 P seems to be a suitable rootstock under possible salinity stress. The findings coincide with the results of other researchers [15,89,90]. Rootstocks derived from *Vitis Berlandieri* × *V. Rupestris* parentage, as 1103 P, could prevent the accumulation of Chloride and Sodium in aerial parts of the vines.

**Author Contributions:** Conceptualization, K.-E.N. and E.Z.; data collection, K.-E.N., S.T. and T.C.; data analysis, K.-E.N.; writing and editing, K.-E.N.; nutrient elements analysis, K.-E.N. and T.C.; review, S.K. and A.A. All authors have read and agreed to the published version of the manuscript.

**Funding:** This research was funded by the Hellenic Foundation for Research and Innovation (H.F.R.I.).

**Institutional Review Board Statement:** Not applicable.

**Informed Consent Statement:** Not applicable.

**Data Availability Statement:** The data presented in this study are available on request from the corresponding author. The data are not publicly available due to privacy reasons.

**Conflicts of Interest:** The authors declare no conflict of interest.

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
