# Peer review of "Effects of Salinity and Rootstock on Nutrient Element Concentrations and Physiology in Own-Rooted or Grafted to 1103 P and 101-14 Mgt Rootstocks of Merlot and Cabernet Franc Grapevine Cultivars under Climate Change"

_sustainability, doi:10.3390/su13052477_

Round 1

Reviewer 1 Report

The scientific paper addresses a topic of great relevance, with applicability in other areas with similar characteristics. The research method and materials used are clear, scientifically substantiated and in accordance with the theme of the paper. The results and discussions are well argued. The only observation is that the bibliography does not contain enough citations from the last 10 years.  

Reviewer 2 Report

Review “Effects of salinity and rootstock on nutrient element concentrations and physiology in own – rooted or grafted to 1103P and 101-14 Mgt rootstocks of Merlot and Cabernet Franc grapevine cultivars under climate change”

Nikolaou K.E., Chatzistathis T., Theocharis S., Argiriou A., Koundouras S., Zioziou E.

Date of review:           25.01.2021

From my view point, the paper quality is sufficiently high for publication in prestigious international scientific journal such as “Sustainability”. This paper provides a comprehensive analysis of how soil salinity can affect grape yields in several regions of the Mediterranean, assuming that soil salinity may be caused by climate change. I suggest only minor changes that should be made by authors before the publication of the manuscript.

Comment 1:    Please remove dot at the end of title.

Comment 2:    Authors: “... global climate models predict an increase in aridity in the next future [1]”.

Question: Where, in what geographical areas? Actually the aridity is projected not around the globe. Make changes accordingly.

Comment 3:    Authors: “During the twentieth century the global mean temperature increased by 0.89 0C [2].”

The reference [2] was published seven years ago. The World Meteorological Organization publishes annual report in which the global mean surface temperature increase is updated. The last report said: “The average global temperature in 2020 is set to be about 1.2 °C above the pre-industrial (1850-1900) level” (see WMO Provisional Report on the State of the Global Climate 2020). Make changes accordingly.

Reviewer 3 Report

Comments to the Authors’

The manuscript entitled “Effects of salinity and rootstock on nutrient element concentrations and physiology in own – rooted or grafted to 1103P and 101-14 Mgt rootstocks of Merlot and Cabernet Franc grapevine cultivars under climate change” contains the scientific novelty. Well written and summarized.

Minor comments:

  • In the introduction, (Vitis vinifera) should be italic. Also, please check throughout the manuscript.
  • In the introduction, in 2 of 18: correct it as “plants of the species Vinifera American vine species”…
  • In the introduction, in 2 of 18: change it as “The photosynthesis inhibition caused by the reduction of stomatal conductance as a direct salinity effect was reported [7]”.
  • In materials and methods: Is it 2.5 L nursery bags or 2 and 5 L nursery bags. Please change accordingly.
  • Please provide a reference for the Hoagland No 2-nutrient solution in materials and methods.
  • Please check for the missing spaces and double spaces in many places.
  • Please check “contents decreased 25,71% in salinity plots.” In the Results section.
  • In Table 3. Please check 44,32 and 1,481? See all the Tables again for such errors.
  • Please add error bars for all the Figures.
  • Results and discussion were well interpreted with figures and tables.

I would recommend the publication of this manuscript after addressing minor changes.
